# FREDNORMER: FREQUENCY DOMAIN NORMALIZATION FOR NON-STATIONARY TIME SERIES FORECASTING

## ABSTRACT

Recent normalization-based methods have shown great success in tackling the distribution shift issue, facilitating non-stationary time series forecasting. Since these methods operate in the time domain, they may fail to fully capture the dynamic patterns that are more apparent in the frequency domain, leading to suboptimal results. This paper first theoretically analyzes how normalization methods affect frequency components. We prove that the current normalization methods that operate in the time domain uniformly scale non-zero frequencies. Thus, they struggle to determine components that contribute to more robust forecasting. Therefore, we propose `FredNormer`, which observes datasets from a frequency perspective and adaptively up-weights the key frequency components. To this end, `FredNormer` consists of two components: a statistical metric that normalizes the input samples based on their frequency stability and a learnable weighting layer that adjusts stability and introduces sample-specific variations. Notably, `FredNormer` is a plug-and-play module, which does not compromise the efficiency compared to existing normalization methods. Extensive experiments show that `FredNormer` improves the averaged MSE of backbone forecasting models by 33.3% and 55.3% on the ETTm2 dataset. Compared to the baseline normalization methods, `FredNormer` achieves 18 top-1 results and 6 top-2 results out of 28 settings. Our code is available at: https://anonymous.4open.science/r/ICLR2025-13956-8F84

## 1 INTRODUCTION

Deep learning models have demonstrated significant success in time series forecasting (Moosavi et al., 2019; Zhou et al., 2021; Wu et al., 2021; M. et al., 2022; Nie et al., 2023; Zhang & Yan, 2023; Liu et al., 2024b). These models aim to extract diverse and informative patterns from historical observations to enhance the accuracy of future time series predictions. To achieve accurate time series predictions, a key challenge is that time series data derived from numerous real-world systems exhibit dynamic and evolving patterns, i.e., a phenomenon known as non-stationarity (Stoica et al., 2005; Box et al., 2015; Xie et al., 2018; Rhif et al., 2019). This characteristic typically results in discrepancies among training, testing, and future unseen data distributions. Consequently, the non-stationary characteristics of time series data necessitate the development of forecasting models that are robust to such temporal shifts in data distribution, while failing to address this challenge often leads to representation degradation and compromised model generalization (Kim et al., 2021; Du et al., 2021; Lu et al., 2023).

A recent fashion to tackle the above-mentioned distribution shift issue is leveraging plug-and-play normalization methods (Kim et al., 2021; Fan et al., 2023; Liu et al., 2023b; Han et al., 2024). These methods typically normalize the input time series to a unified distribution, removing non-stationarity explicitly to reduce discrepancies in data distributions. During the forecasting stage, a denormalization is applied, reintroducing the distribution statistics information to the data. This step ensures the forecasting results are accurate while reflecting the inherent variability and fluctuation in the time series, enhancing generalization. Since they focus on scaling the inputs and outputs, these methods function as "model-friendly modules" that could be easily integrated into various forecasting models without any transformation. However, existing works still face several challenges.

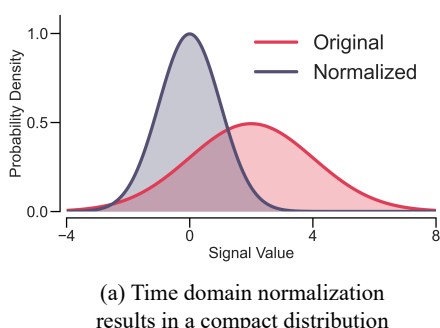 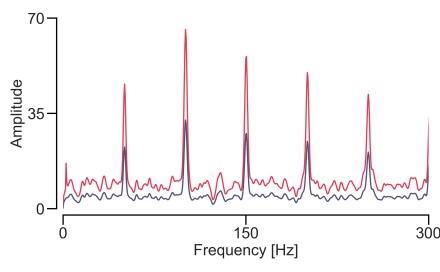

(a) Time domain normalization
results in a compact distribution

(b) Normalization uniformly scale
amplitudes of non-zero frequencies

Figure 1: *How does z-score normalization affect the frequency amplitudes?* (a) Normalization in the time domain compresses the variability of the data, leading to a more compact distribution. (b) The amplitudes of non-zero frequencies are just uniformly scaled after normalization.

(1) Existing methods model *non-stationary in the time domain* that may not fully capture dynamic and evolving patterns in time series. Conceptually, a series of time observations is considered a complex set of waves that varies over time (Proakis & Manolakis, 1996). These temporal variations, manifested as various frequency waves, are intermixed in the real world (He et al., 2023; Wu et al., 2023; Piao et al., 2024). Modeling solely in the time domain struggles to distinguish between different frequency components within superimposed time series, resulting in entangled patterns and sub-optimal performance. Recent works have shown that explicitly modeling frequency can enhance representation quality and forecasting accuracy (Yi et al., 2023; Zhang et al., 2024; Piao et al., 2024). (2) Existing methods primarily rely on *z-score normalization* to rescale the input distribution. However, as illustrated in Figure 1, this normalization applies uniform scaling across all frequency components, which leaves frequency-specific patterns unaltered. Such uniform scaling may reduce distributional differences across frequencies, potentially obscuring important time-invariant features crucial for generalization to unseen time series.

To tackle the challenges above, this paper proposes a novel solution for the distribution shift issue in time series forecasting by modeling the non-stationarity in the *frequency domain*. We first investigate why time-domain normalization does not provide significant benefits for capturing dynamics in frequencies and theoretically prove our findings. We then propose `FredNormer` based on the insight we have gained. Specifically, `FredNormer` learns time-invariant frequency components, termed stable frequencies, to suppress non-stationary for robust forecasting. Finally, we propose a new normalization metric tailored for quantifying the importance or stability of frequencies. Since `FredNormer` operates only on the input time series, it is plug-and-play, making it easily adaptable to various forecasting models and complementary to existing normalization methods.

**Contributions and Novelty.** `FredNormer` tackles the aforementioned two challenges as follows: (1) We first transform the input time series data from the time domain to the spectral domain and extract the statistical significance of frequencies across training sets. We then propose a linear projection to capture data-specific properties, which adjusts the statistical significance used to weight the spectrum. The processed time series data, which carries more stable components while filtering out non-stationary elements, is finally used as input for the forecasting models. (2) To quantify frequency significance, we propose a stability metric based on the well-known Coefficient of Variation (CV) (Aja-Fernández & Alberola-López, 2006; Jalilibal et al., 2021). It computes the ratio between the mean and variance of frequency amplitude, providing a *relative measure (scaling)* of variability rather than the absolute scaling/rescaling (Reed et al., 2002; Abdi, 2010). In summary, our contributions lie in:

- **Theoretical Analysis.** We theoretically analyze how normalization-based methods function in the frequency domain and why they fail to suppress non-stationary frequencies.

- **Novel Problem Formulation.** We are the first to investigate a frequency-based module to tackle the distributional issue in non-stationary time series. Hence, we formulate a key research question in this paper: How can we effectively capture stable frequencies to support robust forecasting?

- **Simple, Efficient, Model-Agnostic Method.** We propose `FredNormer`, which explicitly learns the statistical significance of frequencies using a new stability metric. Only simple linear projection layers with a few parameters need learning and tuning.

We apply `FredNormer` to different forecasting models to validate its effectiveness across various datasets. Overall, in non-stationary datasets, such as Traffic, we improved PatchTST and iTransformer by 33.3% and 55.3%, respectively. `FredNormer` achieved 18 top-1 results and 6 top-2 results out of 28 settings compared to the baselines and outperformed the SOTA normalization method, with speed improvements ranging from 60% to 70% in 16 out of 28 settings.

**Notations.** Vectors and matrices are denoted by lowercase and uppercase boldface letters, respectively (e.g., $\boldsymbol{x}$, $\mathbf{X}$). We consider a training dataset with $N$ labeled samples consisting of $L$ timestamps of past observations and $H$ timestamps of future data. The Discrete Fourier Transform (DFT) of a time series $\mathbf{X}$ is represented by the coefficient matrix $\mathbf{F} \in \mathbb{R}^L$: $\mathbf{F}(k) = \sum_{t=0}^{L-1} x(t)e^{-j\frac{2\pi}{L}kt}$, for $k = 0, 1, \ldots, L-1$. The amplitude matrix $\mathbf{A}$ of a time series $\mathbf{X}$ is defined as $\mathbf{A} = \left[\left|\sum_{t=0}^{L-1} x(t)e^{-j\frac{2\pi}{L}kt}\right|\right]_{k=0}^{L-1}$, where $|\cdot|$ represents the 2-norm of the DFT coefficients. The indicator function $\mathbb{1}\{k = 0\}$ equals 1 if $k$ does not equal 0, and 0 otherwise. $\mu(\cdot)$ and $\delta(\cdot)$ represent the mean and the standard deviation, respectively.

## 2 PROBLEM FORMULATION

In this section, we first define frequency stability and then investigate how normalization in the time domain affects the frequency domain and its influences. Finally, we formulate the research question.

**Definition 1** *(Frequency Stability). Given a training set containing $N$ samples, we define the statistical stability of the $k$-th component as the reciprocal of the Coefficient of Variation $\gamma$:*

$$S(k) := \frac{1}{\gamma(A(k))} = \frac{\mu(A(k))}{\sigma(A(k))} \tag{1}$$

*where $\mu(A(k)) = \frac{1}{N}\sum_{i=1}^{N} A^i(k)$ and $\sigma(A(k)) = \sqrt{\frac{1}{N}\sum_{i=1}^{N}\left(A^i(k) - \mu(A(k))\right)^2}$ are the mean and standard deviation of the amplitude across the training set.*

$S(k)$ measures the statistical significance of each frequency across the dataset. A frequency component with higher stability denotes lower relative variability, i.e., more stable; otherwise, it is considered unstable. All stable components are included in the subset $\mathcal{O}$.

**Definition 2** *(Stable Frequency Subset). Given $K - 1$ non-zero frequencies, a subset $\mathcal{O} = \{1, \ldots, M\}$, where $M \ll K$, contains $M$ components with higher stability $S(k)$.*

**Definition 3** *(Linearity of Fourier Transform.) For any functions $f_1$ and $f_2$, and constants $a$ and $b$, the Fourier Transform $\mathcal{F}$ satisfies:*

$$\mathcal{F}(af_1 + bf_2) = a\mathcal{F}(f_1) + b\mathcal{F}(f_2) \tag{2}$$

Thus, we investigate the variations of $\mathcal{O}$ (Def. 2) before and after z-score normalization in the time domain. Meanwhile, the linearity property (Def. 3) allows us to map this normalization to its corresponding operations in the frequency domain.

**Lemma 1** *Normalization in the time domain uniformly scales non-zero frequency components.*

**Proof 1.** For a normalized time series $\mathbf{X}_z(t) = \frac{\mathbf{X}(t) - \mu(\mathbf{X})}{\sigma(\mathbf{X})}$, a Fourier transform $\mathcal{F}(\cdot)$ is applied by:

$$\mathcal{F}(\mathbf{X}_z(t)) = \mathcal{F}(\frac{\mathbf{X}(t) - \mu(\mathbf{X})}{\sigma(\mathbf{X})}) = \frac{1}{\sigma(\mathbf{X})}\left(\mathcal{F}(\mathbf{X}(t)) - \mu(\mathbf{X})\mathbb{1}\{k = 0\}\right) \quad \text{(by linearity, Def. 3)}$$
$$\tag{3}$$

The left item $\mathcal{F}(\mathbf{X}(t))$ is the Fourier Transform of $\mathbf{X}$. Since $\mu(\mathbf{X})$ is a constant, the resulted Fourier transformation can be represented by $\mu(\mathbf{X})\mathbb{1}\{k = 0\}$, where $\mathbb{1}\{k = 0\}$ is an indicator function, that is, for $k \neq 0$, $\mathbb{1}\{k = 0\} = 0$. Then for the non-zero frequencies ($k \neq 0$),

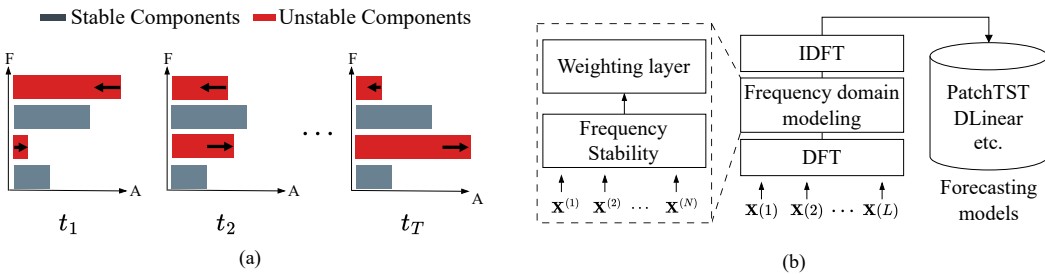

Figure 2: (a) Frequency variations across a sequence of time series samples. The red bar denotes the unstable frequency components, while stable ones are in gray. (b) An overview of FredNormer.

$$\mathcal{F}(\mathbf{X}_z(t)) = \frac{1}{\sigma(\mathbf{X})}\left(\mathcal{F}(\mathbf{X}(t)) - \mu(\mathbf{X}) \cdot 0\right) = \frac{1}{\sigma(\mathbf{X})}\mathcal{F}(\mathbf{X}(t)) \quad \text{for } k \neq 0 \tag{4}$$

Here, the amplitudes of the non-zero frequency components are scaled by $\frac{1}{\sigma(\mathbf{X})}$:

$$|\mathbf{A}_z(k)| = \frac{1}{\sigma(\mathbf{X})}|\mathbf{A}(k)|, \quad \text{for } k \neq 0 \tag{5}$$

Here, $\mathbf{A}$ is the amplitudes of frequencies. Its definition can be found in **Notation** in Section 1. Thus, normalization uniformly scales all non-zero frequency components in the frequency domain. A more detailed proof of Lemma 1 can be found in Appendix A.1.

**Theorem 1** *( The proportion of $\mathcal{O}$ to the spectrum is unchanged after normalization ).*

*The energy proportion of $\mathcal{O}$ in the spectrum is defined by the sum of the amplitudes divided by the energy of the entire spectrum. Then if Lemma 1 holds, we have:*

$$\frac{\sum_{k \in \mathcal{O}} |\mathbf{A}_z(k)|}{\sum_{k=1}^{K-1} |\mathbf{A}_z(k)|} = \frac{\sum_{k \in \mathcal{O}} |\mathbf{A}(k)|}{\sum_{k=1}^{K-1} |\mathbf{A}(k)|} \tag{6}$$

*The left and right items represent the ratio after and before normalization, respectively. As shown, this ratio remains the same. A proof of Theorem 1 can be found in Appendix A.2.*

**Remark.** Since Theorem 1 holds, the normalization operation keeps the proportion unchanged. The stable components in $\mathcal{O}$ are often intermixed with unstable components in the time series data. This makes forecasting models struggle to distinguish between stable and unstable components, resulting in entangled patterns and sub-optimal performance (Elvander & Jakobsson, 2020; Piao et al., 2024). This motivates us to increase the weights of stable components and suppress unstable ones. In this paper, we aim to learn a frequency-based method for assigning higher weights to stable components and improving forecasting performance for future data.

**Problem 1** *(Enhancing stable components for better generalization). Given a time series dataset with stable frequency components $\mathcal{O}$, our goal is to develop a module $f(\cdot)$ that dynamically adjusts $\mathcal{O}$, or the amplitudes, increasing the energy proportion to the spectrum:*

$$f(\mathbf{A}(k)) := w(k) \cdot \mathbf{A}(k), \quad \text{where } w(k_1) > w(k_2), \text{ for } k_1 \in \mathcal{O} \text{ and } k_2 \notin \mathcal{O}.$$

Here $w(\cdot)$ represents the weighting function. In this paper, we assume $w(\cdot)$ should have two properties: ( i ) extracting the statistical significance of frequencies across samples in the training set; ( ii ) capturing data-specific properties to adjust the statistical significance.

## 3 METHOD: FREDNORMER

In this section, we present FredNormer (as shown in Figure 2), which consists of two components: ( i ) a frequency stability measure, and ( ii ) a frequency stability weighting layer, followed by:

| **Algorithm 1** Frequency Stability Measure | **Algorithm 2** Frequency Stability Weighting |
|---|---|
| 1: **input:** $train\_loader$ contains $N$ samples | 1: **input:** stability $S$ and sample $X$ |
| 2: **for** each sample $X^{(i)}$ in $train\_loader$ **do** | 2: $X \leftarrow$ 1D-difference($X$) |
| 3: $\quad A^{(i)} \leftarrow |\text{FFT}(X^{(i)})|$ | 3: $F \leftarrow \text{DFT}(X)$ |
| 4: $\quad sum(A) \leftarrow sum(A) + \sum_k A^{(i)}(k)$ | 4: **for** $k = 1$ to $K - 1$ **do** |
| 5: $\quad sum(A^2) \leftarrow sum(A^2) + \sum_k \left(A^{(i)}(k)\right)^2$ | 5: $\quad W_r(k), B_r(k) \leftarrow \text{linear\_r}(S)$ |
| 6: **end for** | 6: $\quad W_i(k), B_i(k) \leftarrow \text{linear\_i}(S)$ |
| 7: $\mu(A) \leftarrow sum(A)/N$ | 7: $\quad F_{real}(k) \leftarrow F.real(k) \circ W_r(k) + B_r(k)$ |
| 8: $\sigma(A) \leftarrow \sqrt{(sum(A^2)/N - \mu(A)^2 + 10^{-5})}$ | 8: $\quad F_{imag}(k) \leftarrow F.imag(k) \circ W_i(k) + B_i(k)$ |
| | 9: **end for** |
| 9: $S \leftarrow \mu(A)/(\sigma(A) + 10^{-5})$ | 10: $F_{weighted} \leftarrow \text{complex}(F_{real}, F_{imag})$ |
| | 11: $X' \leftarrow \text{IDFT}(F_{weighted}).real$ |
| 10: **return** $S$ | 12: **return** $X'$ |

- First, we compute the frequency stability, defined in Definition 1, for a given time series dataset. The output is a **statistical measure** that can be tuned for different data scenarios.

- Second, the input time series is transformed into the frequency spectrum using the DFT.

- Third, during the training phase, a learnable linear projection adjusts the frequency stability measure for the spectrum to introduce **sample-specific variation**, increasing distributional diversity.

- Finally, `FredNormer` transforms the adjusted frequency spectrum back into the time domain using the Inverse-DFT (IDFT) that serves as input for subsequent various forecasting models.

The detailed workflows, including two key components, are presented in Algorithms 1 and 2.

**Frequency Stability Measure.** Given a training set contains $N$ time series samples $\mathcal{X} = \{\mathbf{X}^{(i)}\}_{i=1}^N$, we first apply the DFT to each sample $\mathbf{X}^{(i)} \in \mathbb{R}^{L \times C}$ to transform it into $\mathbf{A}^{(i)} \in \mathbb{R}^{K \times C}$. Here, $K$ is the number of frequency components, and $C$ denotes the number of channels. The frequency stability measure $S(k) \in \mathbb{R}^{K \times C}$ is then applied:

$$S(k) = \frac{\mu(\mathbf{A}(k))}{\sigma(\mathbf{A}(k))} = \frac{\frac{1}{N} \sum_{i=1}^N \mathbf{A}(k)^{(i)}}{\sqrt{\frac{1}{N} \sum_{i=1}^N \left(\mathbf{A}(k)^{(i)} - \mu(\mathbf{A}(k))\right)^2}} \tag{7}$$

where $\mu(\mathbf{A}(k)) \in \mathbb{R}^{K \times C}$ and $\sigma(\mathbf{A}(k)) \in \mathbb{R}^{K \times C}$. A larger $\mu(\cdot)$ indicates a higher energy proportion in the spectrum, while a higher $\sigma(\cdot)$ denotes greater sample dispersion. $S$ has two key properties:

- It captures the distribution of each frequency component across the entire training set. This allows the forecasting model to learn the *overall stability* of each component across different samples.

- $S$ is a dimensionless measure that allows for a fair comparison between different frequency components, thus avoiding uniform frequency scaling, as defined in Theorem 1.

**Frequency Stability Weighting Layer.** Given the input multivariate time series data $\mathbf{X} \in \mathbb{R}^{L \times C}$, we first apply a 1-D differencing operation to smooth the data and then transform $\mathbf{X}$ into the spectrum, decomposing it into the DFT coefficients:

$$\mathbf{F}(k, c) = \sum_{l=0}^{L-1} \Delta(\mathbf{X}(l, c)) \cdot e^{-2\pi i k n / L}, \quad k = 0, 1, \ldots, K - 1 \tag{8}$$

where $\Delta(\cdot)$ denotes the differencing operation, and two matrices are produced, representing the real and imaginary parts, $(\mathbf{F}_r, \mathbf{F}_i)$, formulated as:

$$\mathbf{F}_r(k, c) = \sum_{l=0}^{L-1} \mathbf{X}(l, c) \cdot \cos\left(\frac{2\pi k n}{L}\right) \quad \mathbf{F}_i(k, c) = -\sum_{l=0}^{L-1} \mathbf{X}(l, c) \cdot \sin\left(\frac{2\pi k n}{L}\right) \tag{9}$$

Next, we apply two linear projections to $\mathbf{S}$ to the real and imaginary parts separately:

$$\mathbf{F}'_r = \mathbf{F}_r \odot (\mathbf{S} \times \mathbf{W}_r + \mathbf{B}_r), \quad \mathbf{F}'_i = \mathbf{F}_i \odot (\mathbf{S} \times \mathbf{W}_i + \mathbf{B}_i), \tag{10}$$

where $\mathbf{W}_r$ and $\mathbf{W}_i \in \mathbb{R}^{K \times 1}$ denote weight matrices for the $\mathbf{F}_r$ and $\mathbf{F}_i$, respectively. $\mathbf{B}_r, \mathbf{B}_i \in \mathbb{R}^K$ are bias vectors, and $\odot$ denotes Hadamard product. We handle the real and imaginary parts with separate networks because they correspond to different basis functions, allowing us to capture diverse temporal dynamics (Zhang et al., 2024; Piao et al., 2024). Finally, we transform $(\mathbf{F}_r, \mathbf{F}_i)$, with enhanced stable frequency components, back into the time domain by:

$$\mathbf{X}'(l, c) = \sum_{k=0}^{K-1} \left( \mathbf{F}'_r(k, c) \cdot \cos\left(\frac{2\pi k l}{L}\right) - \mathbf{F}'_i(k, c) \cdot \sin\left(\frac{2\pi k l}{L}\right) \right) \tag{11}$$

where $\mathbf{X}'$, with the same size as $\mathbf{X} \in \mathbb{R}^{L \times C}$, serves as the input for various forecasting models.

## 4 EXPERIMENTS

Table 1: Multivariate forecasting results (average) with forecasting lengths $H \in \{96, 192, 336, 720\}$ for all datasets and fixed input sequence length $L = 96$.

| Models | PatchTST (Nie et al., 2023) | | | | iTransformer (Liu et al., 2024b) | | | |
|---|---|---|---|---|---|---|---|---|
| Methods | + Ours | | Ori | | + Ours | | Ori | |
| Metric | MSE | MAE | MSE | MAE | MSE | MAE | MSE | MAE |
| Electricity | **0.197** ± 0.027 | **0.296** ± 0.033 | 0.218 ± 0.31 | 0.307 ± 0.032 | **0.169** ± 0.035 | **0.262** ± 0.041 | 0.179 ± 0.28 | 0.279 ± 0.046 |
| ETTh1 | **0.438** ± 0.024 | **0.437** ± 0.035 | 0.480 ± 0.037 | 0.481 ± 0.031 | **0.445** ± 0.017 | **0.443** ± 0.026 | 0.511 ± 0.033 | 0.496 ± 0.036 |
| ETTh2 | **0.379** ± 0.032 | **0.380** ± 0.038 | 0.604 ± 0.130 | 0.524 ± 0.027 | **0.376** ± 0.041 | **0.400** ± 0.057 | 0.813 ± 0.134 | 0.666 ± 0.072 |
| ETTm1 | **0.390** ± 0.027 | **0.398** ± 0.025 | 0.419 ± 0.055 | 0.432 ± 0.047 | **0.396** ± 0.026 | **0.406** ± 0.056 | 0.447 ± 0.026 | 0.457 ± 0.061 |
| ETTm2 | **0.280** ± 0.032 | **0.325** ± 0.031 | 0.420 ± 0.035 | 0.424 ± 0.044 | **0.283** ± 0.020 | **0.327** ± 0.026 | 0.633 ± 0.055 | 0.489 ± 0.041 |
| Traffic | **0.427** ± 0.029 | **0.285** ± 0.025 | 0.619 ± 0.077 | 0.365 ± 0.029 | **0.424** ± 0.031 | **0.282** ± 0.027 | 0.576 ± 0.069 | 0.372 ± 0.035 |
| Weather | **0.251** ± 0.019 | **0.276** ± 0.017 | 0.255 ± 0.021 | 0.312 ± 0.031 | **0.246** ± 0.023 | **0.274** ± 0.017 | 0.274 ± 0.029 | 0.320 ± 0.041 |

### 4.1 EXPERIMENTAL SETTINGS

**Datasets.** We conducted experiments on seven public time series datasets, including Weather, four ETT repositories (ETTh1, ETTh2, ETTm1, ETTm2), Electricity (ECL), and Traffic dataset. For example, the Electricity dataset includes the hourly electricity consumption record in 321 households. All datasets are available in (Liu et al., 2024b) .

**Baselines.** We selected RevIN (Kim et al., 2021) and SAN (Liu et al., 2023b) as our baselines.

- RevIN is widely used as a fundamental module in various forecasting models, including PatchTST (Nie et al., 2023), Crossformer (Zhang & Yan, 2023), iTransformer (Liu et al., 2024b), Fredformer (Piao et al., 2024), among others (Wang et al., 2024).
- SAN (Liu et al., 2023b) is the new state-of-the-art (SOTA) method, outperforming several non-stationary forecasting modules (Kim et al., 2021; Fan et al., 2023) and models (Liu et al., 2022b).

**Backbones and Setup.** For fair comparisons, we selected three forecasting models, including DLinear (Zeng et al., 2023), PatchTST (Nie et al., 2023), and iTransformer (Liu et al., 2024b), as the backbone, and deployed all three modules (`FredNormer`, RevIN, and SAN) for evaluation. DLinear is a simple yet efficient forecasting model with an architecture solely involving MLPs. PatchTST and iTransformer are two well-known Transformer methods that frequently serve as baselines in various forecasting research (Liu et al., 2024b;a; Piao et al., 2024; Zhang et al., 2024). We followed the implementation and setup provided in (Liu et al., 2023b)[1] and (Liu et al., 2024b)[2].

**Experiments Details.** We used mean squared error (MSE) and mean absolute error (MAE) as the evaluation metrics, where lower values indicate better performance. All experiments were implemented on a single NVIDIA RTX A6000 48GB GPU with CUDA V12.4. More details of the datasets are in Appendix B.1, the baselines are in B.2, the backbones and setup are in B.3, and other details of the experiments are in B.4.

---

[1] https://github.com/icantnamemyself/SAN
[2] https://github.com/thuml/Time-Series-Library

Table 2: Multivariate forecasting results (average) with $H \in \{96, 192, 336, 720\}$ for all datasets and fixed input sequence length $L = 96$. The **best** and second best results are highlighted. Ours* represents the results where both `FredNormer` and SAN are used in the backbones.

| Models | MLP-based (DLinear(Zeng et al., 2023)) | | | | | | | | Transformer-based (iTransformer(Liu et al., 2024b)) | | | | | | | |
| --- | --- | --- | --- | --- | --- | --- | --- | --- | --- | --- | --- | --- | --- | --- | --- | --- |
| Methods | + Ours* | | + Ours | | +SAN | | + RevIN | | + Ours* | | + Ours | | +SAN | | + RevIN | |
| Metric | MSE | MAE | MSE | MAE | MSE | MAE | MSE | MAE | MSE | MAE | MSE | MAE | MSE | MAE | MSE | MAE |
| Electricity | **0.161** | **0.257** | 0.168 | 0.262 | 0.163 | 0.260 | 0.225 | 0.316 | 0.175 | 0.273 | **0.169** | **0.262** | 0.195 | 0.283 | 0.205 | 0.272 |
| ETTh1 | 0.413 | 0.424 | **0.407** | **0.419** | 0.421 | 0.427 | 0.460 | 0.456 | 0.455 | 0.449 | **0.445** | **0.443** | 0.466 | 0.455 | 0.463 | 0.452 |
| ETTh2 | 0.339 | **0.384** | **0.337** | **0.384** | 0.342 | 0.387 | 0.561 | 0.518 | 0.378 | 0.408 | **0.376** | **0.400** | 0.392 | 0.413 | 0.385 | 0.412 |
| ETTm1 | **0.341** | **0.372** | 0.357 | 0.375 | 0.344 | 0.376 | 0.413 | 0.407 | **0.389** | **0.398** | 0.396 | 0.406 | 0.401 | 0.406 | 0.406 | 0.410 |
| ETTm2 | **0.255** | 0.316 | 0.256 | **0.313** | 0.260 | 0.318 | 0.350 | 0.413 | 0.285 | 0.334 | **0.283** | **0.327** | 0.287 | 0.336 | 0.294 | 0.337 |
| Traffic | 0.432 | 0.297 | **0.430** | **0.291** | 0.440 | 0.302 | 0.624 | 0.383 | 0.459 | 0.313 | **0.424** | **0.282** | 0.520 | 0.341 | 0.430 | 0.312 |
| Weather | **0.224** | **0.271** | 0.237 | 0.272 | 0.227 | 0.276 | 0.265 | 0.317 | **0.244** | 0.282 | 0.246 | **0.274** | 0.247 | 0.291 | 0.263 | 0.288 |
| Count | **4** | **4** | 3 | **4** | 0 | 0 | 0 | 0 | 2 | 1 | **5** | **6** | 0 | 0 | 0 | 0 |

Table 3: Detailed results of three selected datasets with $H \in \{96, 192, 336, 720\}$ and input sequence length $L \in \{96, 336\}$. The **best** and second best results are highlighted. Ours* represents the results where both `FredNormer` and SAN are used in the backbones.

| Models | | DLinear (Zeng et al., 2023) | | | | | | | | iTransformer (Liu et al., 2024b) | | | | | | | |
| --- | --- | --- | --- | --- | --- | --- | --- | --- | --- | --- | --- | --- | --- | --- | --- | --- | --- |
| Methods | | + Ours* | | + Ours | | + SAN | | + RevIN | | + Ours* | | + Ours | | + SAN | | + RevIN | |
| Metric | | MSE | MAE | MSE | MAE | MSE | MAE | MSE | MAE | MSE | MAE | MSE | MAE | MSE | MAE | MSE | MAE |
| Electricity | 96 | **0.135** | **0.230** | 0.140 | 0.237 | 0.137 | 0.234 | 0.210 | 0.278 | 0.145 | 0.244 | **0.143** | **0.237** | 0.171 | 0.262 | 0.152 | 0.251 |
| | 192 | **0.149** | **0.245** | 0.155 | 0.249 | 0.151 | 0.247 | 0.210 | 0.304 | 0.169 | 0.266 | **0.159** | **0.252** | 0.180 | 0.270 | 0.264 | 0.255 |
| | 336 | **0.165** | **0.262** | 0.171 | 0.267 | 0.166 | 0.264 | 0.223 | 0.309 | 0.178 | 0.271 | **0.172** | **0.266** | 0.194 | 0.284 | 0.180 | 0.272 |
| | 720 | **0.198** | **0.291** | 0.208 | 0.298 | 0.201 | 0.295 | 0.257 | 0.349 | 0.210 | 0.311 | **0.205** | **0.295** | 0.237 | 0.319 | 0.227 | 0.312 |
| ETTh1 | 96 | 0.375 | 0.398 | **0.371** | **0.392** | 0.383 | 0.399 | 0.396 | 0.410 | **0.380** | **0.400** | 0.389 | 0.404 | 0.398 | 0.411 | 0.394 | 0.409 |
| | 192 | 0.410 | 0.417 | **0.404** | **0.412** | 0.419 | 0.419 | 0.445 | 0.440 | **0.429** | **0.427** | 0.447 | 0.440 | 0.438 | 0.435 | 0.460 | 0.449 |
| | 336 | 0.430 | 0.427 | **0.426** | **0.426** | 0.437 | 0.432 | 0.487 | 0.465 | **0.479** | **0.451** | 0.492 | 0.463 | 0.481 | 0.456 | 0.501 | 0.475 |
| | 720 | 0.437 | 0.455 | **0.428** | **0.448** | 0.446 | 0.459 | 0.512 | 0.510 | **0.491** | **0.471** | 0.496 | 0.482 | 0.528 | 0.502 | 0.521 | 0.504 |
| Traffic | 96 | 0.410 | 0.286 | **0.408** | **0.277** | 0.412 | 0.288 | 0.648 | 0.396 | 0.400 | 0.271 | **0.394** | **0.268** | 0.502 | 0.329 | 0.401 | 0.277 |
| | 192 | 0.427 | 0.288 | **0.422** | **0.283** | 0.429 | 0.297 | 0.598 | 0.370 | 0.470 | 0.319 | **0.413** | **0.277** | 0.490 | 0.331 | 0.421 | 0.282 |
| | 336 | 0.439 | 0.305 | **0.436** | **0.295** | 0.445 | 0.306 | 0.605 | 0.373 | 0.489 | 0.333 | **0.428** | **0.283** | 0.512 | 0.341 | 0.434 | 0.389 |
| | 720 | **0.454** | **0.311** | 0.455 | **0.311** | 0.474 | 0.319 | 0.645 | 0.395 | 0.478 | 0.330 | **0.463** | **0.301** | 0.576 | 0.364 | 0.465 | 0.302 |

## 4.2 RESULTS

**Main Results.** Table 1 presents the overall forecasting results using iTransformer and PatchTST as the backbone across seven datasets. We set the forecasting lengths as $H \in \{96, 192, 336, 720\}$, with the input sequence length $L = 96$. Here, we present the averaged MSE and MAE over four forecasting lengths. We combine our module with a z-score normalization-denormalization operation in all experiments. Obviously, applying `FredNormer` consistently improved the performance of the backbone models across all datasets, as shown in all bold results. More importantly, in datasets with complex frequency characteristics, such as ETTm2, `FredNormer` improves PatchTST and iTransformer by 33.3% ($0.420 \rightarrow 0.280$) and 55.3% ($0.633 \rightarrow 0.283$), respectively. This improvement is attributed to giving higher weights to stable frequency components, allowing them to dominate the adjusted input time series.

**Comparison with Baseline Normalization Methods.** Table 2 presents the average comparison results between `FredNormer` and the baseline normalization methods, i.e., RevIN and SAN. We use the same parameters and forecasting length as in Table 1. For iTransformer, the input sequence length is $L = 96$, and $L = 336$ for DLinear. As shown, `FredNormer` (denoted as "Ours" in the table) achieves 18 top-1 results and 6 top-2 results out of 28 settings. For instance, n the ETTh1 dataset, `FredNormer` improves the MSE values for DLinear and iTransformer to $0.407$ and $0.445$, outperforming RevIN ($0.460$ and $0.463$) and SAN ($0.421$ and $0.466$). Similarly, in the Traffic dataset, `FredNormer` improves the MSE value to $0.430$, compared to RevIN ($0.624$) and SAN ($0.440$). Notably, as we highlighted in Section 1, one purpose of `FredNormer` is to complement existing normalization methods in the frequency domain. Here, **Ours*** represents the incorporation of SAN into the backbones, which further improves the second-best results (underlined in the table) to the best. For example, in the ETTh1 dataset with $H = 96$ and 192 on the iTransformer backbone, the results improved from $0.389$ and $0.447$ to $0.380$ and $0.429$. Table 3 shows detailed results on three selected datasets, with all results provided in Appendix C.1.

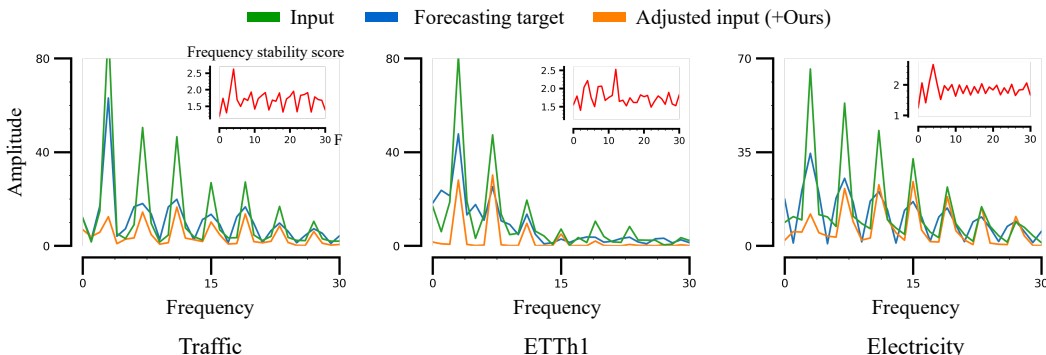

Figure 3: Visualization of input sequences before and after applying `FredNormer` on the Traffic, ETTh1, and ETTh2 datasets. The green line shows the input data, the blue line represents the forecasting target, and the orange line illustrates the input data generated by `FredNormer`. The red line represents the frequency stability measure of each dataset.

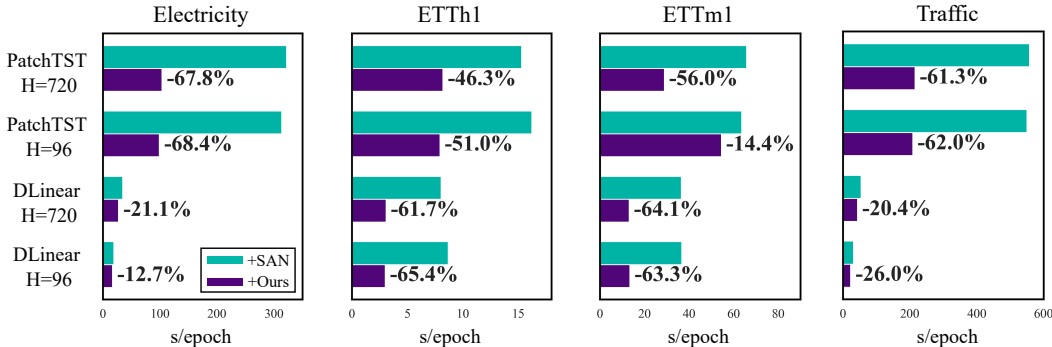

Figure 4: Comparison of running times (s/epoch) between `FredNormer` and SAN on DLinear and PatchTST. Forecasting lengths $H \in \{96, 720\}$ for all datasets and input sequence length $L = 96$.

**Frequency Stability Measure Analysis.** Figure 3 presents an empirical analysis visualizing the frequency stability measures across three datasets. The green line and the blue line represent the amplitudes (FFT outputs) of the input series and forecasting target, respectively. The orange line shows the adjusted input series using `FredNormer`. The red line represents the frequency stability measures, assigning higher weights to components with significant fluctuations appearing in both the input series and forecasting target, while down-weighting components with low amplitudes. Meanwhile, we observe that these measures are adaptive to different datasets. Interestingly, the fourth-shaped frequency component exhibits relatively consistent amplitudes, and although its amplitude value is lower, our metric assigns it a higher weight. In the Electricity dataset, there is a degeneration trend in amplitude, but due to high consistency, the higher frequency components maintain a higher weight.

**Running Time.** Figure 3 presents the running time results for `FredNormer` and the SOTA SAN across four datasets. Full computation results for all seven datasets are available in Appendix C.2. We compare the computation time for both methods at the shortest forecasting length ($H = 96$) and the longest ($H = 720$). The results show the average computation time (in seconds per epoch) using DLinear and PatchTST as backbone models. `FredNormer` consistently outperforms SAN across all datasets. Notably, we achieved improvements of 60% to 70% in 16 out of 28 settings (see Appendix C.2). These improvements are primarily due to the fact that `FredNormer` only utilizes DFT and linear layers during the training phase, minimizing its impact on computation time.

**Frequency Measure Ablation.** Table 4 shows the results of replacing the frequency stability measure with two alternative filters. Our metric can be viewed as a filter that learns statistical measures across datasets and then applies filtering. Since `FredNormer` may focus on higher frequency components, we first compare it to a low-pass filter. Additionally, we include frequency random

Table 4: Results for each setting in the ablation study with forecasting lengths $H \in \{96, 720\}$ for all datasets. The **best** results are highlighted.

| Dataset | | ETTh1 | | ETTm1 | | Weather | |
|---------|-----------|-------|-------|-------|-------|---------|-------|
| Length | | 96 | 720 | 96 | 720 | 96 | 720 |
| Ours | DLinear | **0.371** ± 0.032 | **0.428** ± 0.039 | **0.299** ± 0.021 | **0.425** ± 0.032 | **0.162** ± 0.045 | **0.325** ± 0.041 |
| | iTransformer | **0.389** ± 0.023 | **0.496** ± 0.034 | **0.330** ± 0.035 | **0.475** ± 0.043 | **0.162** ± 0.044 | **0.340** ± 0.036 |
| Low-pass | DLinear | 0.379 ± 0.025 | 0.435 ± 0.036 | 0.305 ± 0.022 | 0.431 ± 0.031 | 0.170 ± 0.089 | 0.337 ± 0.024 |
| | iTransformer | 0.401 ± 0.017 | 0.502 ± 0.029 | 0.335 ± 0.033 | 0.483 ± 0.047 | 0.171 ± 0.055 | 0.356 ± 0.039 |
| Random | DLinear | 0.393 ± 0.066 | 0.440 ± 0.087 | 0.308 ± 0.053 | 0.429 ± 0.079 | 0.171 ± 0.102 | 0.345 ± 0.065 |
| | iTransformer | 0.407 ± 0.048 | 0.533 ± 0.055 | 0.339 ± 0.055 | 0.487 ± 0.061 | 0.172 ± 0.088 | 0.372 ± 0.059 |

selection, a selective method proposed by FEDformer (Zhou et al., 2022b). The results show that our frequency stability score consistently achieved the best accuracy, demonstrating that extracting stable features from the spectrum helps the model learn consistent patterns.

## 5 RELATED WORKS.

**Time Series Forecasting.** Transformers have demonstrated significant success in time series forecasting (Nie et al., 2023; Zhang & Yan, 2023; Jiang et al., 2023), with early works focusing on improving computational efficiency (Li et al., 2019; Beltagy et al., 2020; Zhou et al., 2021; Liu et al., 2022a) and recent works focusing on modeling temporal dependencies (Nie et al., 2023; Liu et al., 2024b). Some other works argue that understanding cross-channel correlations is critical for accurate forecasting. Approaches utilizing Graph Neural Networks (GNNs) (Wu et al., 2020; Cao et al., 2021) and channel-wise Transformer-based frameworks like Crossformer (Zhang & Yan, 2023) and iTransformer (Liu et al., 2024b) captures channel-wise dependencies for forecasting.

**Normalization-based Methods.** RevIN (Kim et al., 2021) is an innovative normalization work for suppressing non-stationary. It employs z-score normalization (i.e., mean of 0 and variance of 1) for input samples, then denormalizes the outputs using the same statistics. Dish-TS (Fan et al., 2023) utilizes learned mean and variance for denormalization. SAN (Liu et al., 2023b) models non-stationary in a set of fine-grained sub-series and proposes an additional loss function to predict their statistics. Instead of mean and variance, SIN (Han et al., 2024) proposes an independent neural network to learn features as the objectives of normalization and denormalization adaptively. However, existing methods focus on modeling statistical variations in the time domain (Liu et al., 2024a).

**Frequency Analysis Methods.** Incorporating frequency information into models can improve forecasting (Wu et al., 2021; Zhou et al., 2022b; Wang et al., 2022; Wu et al., 2023; Yi et al., 2023). Recent study (Piao et al., 2024) recognizes a learning bias issue of frequency in the time domain modeling. CoST (Woo et al., 2022) proposes a pre-training strategy to learn time-invariant representations in the frequency domain. FiLM (Zhou et al., 2022a) employs a low-rank approximation method to extract informative frequencies. Koopa (Liu et al., 2023a) introduces Koopman dynamics (Koopman, 1931) to learn time-invariant frequency features. While promising, existing methods are costly and architecture-specific, limiting their generalization to other forecasting models.

## 6 CONCLUSION

This paper theoretically analyzed the effect of normalization methods on frequency components. We proved that current time-domain normalization methods uniformly scale non-zero frequencies, making it difficult to identify components that contribute to robust forecasting. To address this, we proposed `FredNormer`, which analyzed datasets from a frequency perspective and adaptively up-weighted key frequency components. `FredNormer` consisted of two components: a statistical metric that normalized input samples based on frequency stability, and a learnable weighting layer that adjusted stability and introduced sample-specific variations. Notably, `FredNormer` was a plug-and-play module that maintained efficiency compared to existing normalization methods. Extensive experiments showed that `FredNormer` reduced the average MSE of backbone forecasting models by 33.3% and 55.3% on the ETTm2 dataset. Compared to baseline normalization methods, `FredNormer` achieved 18 top-1 and 6 top-2 results out of 28 settings.

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

# `FredNormer`: Frequency Domain Normalization for Non-stationary Time Series Forecasting
———— Appendix ————

## CONTENTS

## A  PROOFS

We now give the proof of Lemma and theorem in Sec.2. We begin with the proof of Lemma 1.

### A.1  PROOF OF LEMMA 1

Now, we provide proof of normalization in the time domain uniformly scales non-zero frequency components.

**Lemma 1** *(Time Domain Normalization Equates to Uniform Scaling in the Frequency Domain)*

*Normalization in the time domain uniformly scales non-zero frequency components.*

**Proof:**

Let $\mathbf{X}(t)$ be a time series with mean $\mu(\mathbf{X})$ and standard deviation $\sigma(\mathbf{X})$. After applying z-score normalization, we obtain:

$$\mathbf{X}_z(t) = \frac{\mathbf{X}(t) - \mu(\mathbf{X})}{\sigma(\mathbf{X})} \tag{12}$$

Applying the Fourier Transform $\mathcal{F}$ to $\mathbf{X}_z(t)$ and using the linearity property (Definition 3), we have:

$$\mathcal{F}[\mathbf{X}_z(t)] = \mathcal{F}\left[\frac{\mathbf{X}(t) - \mu(\mathbf{X})}{\sigma(\mathbf{X})}\right] \tag{13}$$

$$= \frac{1}{\sigma(\mathbf{X})}\mathcal{F}\left[\mathbf{X}(t) - \mu(\mathbf{X})\right] \tag{14}$$

$$= \frac{1}{\sigma(\mathbf{X})}\left(\mathcal{F}[\mathbf{X}(t)] - \mu(\mathbf{X})\mathcal{F}[1]\right) \tag{15}$$

Since $\mathcal{F}[1]$ is the Fourier Transform of the constant function 1, which equals $\mathbb{1}\{k = 0\}$ (the indicator function):

$$\mathcal{F}[1] = \mathbb{1}\{k = 0\} \tag{16}$$

The indicator function $\mathbb{1}\{k = 0\}$ is nonzero only at $k = 0$. Therefore, for non-zero frequency components ($k \neq 0$):

$$\mathcal{F}[\mathbf{X}_z(t)] = \frac{1}{\sigma(\mathbf{X})}\left(\mathcal{F}[\mathbf{X}(t)] - \mu(\mathbf{X})\mathbb{1}\{k = 0\}\right) \tag{17}$$

$$= \frac{1}{\sigma(\mathbf{X})}\mathcal{F}[\mathbf{X}(t)] \quad \text{for } k \neq 0 \tag{18}$$

This shows that, for $k \neq 0$, the Fourier Transform of the normalized signal is the original Fourier Transform scaled by $\frac{1}{\sigma(\mathbf{X})}$.

Therefore, the amplitudes satisfy:

$$|\mathbf{A}_z(k)| = \frac{1}{\sigma(\mathbf{X})}|\mathbf{A}(k)|, \quad \text{for } k \neq 0 \tag{19}$$

Here, $|\mathbf{A}(k)|$ and $|\mathbf{A}_z(k)|$ are the amplitudes before and after normalization, respectively.

Thus, normalization in the time domain uniformly scales all non-zero frequency components by the factor $\frac{1}{\sigma(\mathbf{X})}$.

A.2 PROOF OF THEOREM 1

Now, based on the proof of lemma 1, we provide proof of normalization Preserves the Proportion of $\mathcal{O}$ in the Spectrum.

**Theorem 1** *(Normalization Preserves the Proportion of $\mathcal{O}$ in the Spectrum)*

*The energy proportion of $\mathcal{O}$ in the spectrum is defined as the sum of its amplitudes divided by the sum of amplitudes of the entire spectrum. If Lemma 1 holds, then:*

$$\frac{\sum_{k \in \mathcal{O}} |\mathbf{A}_z(k)|}{\sum_{k=1}^{K-1} |\mathbf{A}_z(k)|} = \frac{\sum_{k \in \mathcal{O}} |\mathbf{A}(k)|}{\sum_{k=1}^{K-1} |\mathbf{A}(k)|} \tag{20}$$

*The left side represents the ratio after normalization, and the right side represents the ratio before normalization. As shown, this ratio remains unchanged.*

**Proof:**

From Lemma 1, for all non-zero frequencies $k \neq 0$, we have:

$$|\mathbf{A}_z(k)| = \frac{1}{\sigma(\mathbf{X})}|\mathbf{A}(k)| \tag{21}$$

Therefore, the sum of amplitudes in $\mathcal{O}$ after normalization is:

$$\sum_{k \in \mathcal{O}} |\mathbf{A}_z(k)| = \sum_{k \in \mathcal{O}} \frac{1}{\sigma(\mathbf{X})}|\mathbf{A}(k)| \tag{22}$$

$$= \frac{1}{\sigma(\mathbf{X})} \sum_{k \in \mathcal{O}} |\mathbf{A}(k)| \tag{23}$$

Similarly, the sum of amplitudes of the entire spectrum (excluding $k = 0$) after normalization is:

$$\sum_{k=1}^{K-1} |\mathbf{A}_z(k)| = \sum_{k=1}^{K-1} \frac{1}{\sigma(\mathbf{X})}|\mathbf{A}(k)| \tag{24}$$

$$= \frac{1}{\sigma(\mathbf{X})} \sum_{k=1}^{K-1} |\mathbf{A}(k)| \tag{25}$$

Calculating the energy proportion after normalization:

$$\frac{\sum_{k \in \mathcal{O}} |\mathbf{A}_z(k)|}{\sum_{k=1}^{K-1} |\mathbf{A}_z(k)|} = \frac{\frac{1}{\sigma(\mathbf{X})} \sum_{k \in \mathcal{O}} |\mathbf{A}(k)|}{\frac{1}{\sigma(\mathbf{X})} \sum_{k=1}^{K-1} |\mathbf{A}(k)|} \tag{26}$$

$$= \frac{\sum_{k \in \mathcal{O}} |\mathbf{A}(k)|}{\sum_{k=1}^{K-1} |\mathbf{A}(k)|} \tag{27}$$

The scaling factor $\frac{1}{\sigma(\mathbf{X})}$ cancels out in the numerator and denominator. Therefore, the energy proportion of $\mathcal{O}$ remains the same after normalization.

# B    DETAILS OF THE EXPERIMENTS

## B.1    DETAILS OF THE DATASETS.

Weather contains 21 channels (e.g., temperature and humidity) and is recorded every 10 minutes in 2020. ETT (Zhou et al., 2021) (Electricity Transformer Temperature) consists of two hourly-level datasets (ETTh1, ETTh2) and two 15-minute-level datasets (ETTm1, ETTm2). Electricity (Lai et al., 2018a), from the UCI Machine Learning Repository and preprocessed by, is composed of the hourly electricity consumption of 321 clients in kWh from 2012 to 2014. Solar-Energy (Lai et al., 2018b) records the solar power production of 137 PV plants in 2006, sampled every 10 minutes. Traffic contains hourly road occupancy rates measured by 862 sensors on San Francisco Bay area freeways from January 2015 to December 2016. More details of these datasets can be found in Table.5.

## B.2    DETAILS OF THE BASELINES

**Reversible Instance Normalization.** Reversible Instance Normalization (Revin) normalizes each input sample using z-score normalization while preserving the original mean and variance. Revin

Table 5: Overview of Datasets

| Dataset | Source | Resolution | Channels | Time Range |
|---------|--------|------------|----------|------------|
| Weather | Autoformer(Wu et al., 2021) | Every 10 minutes | 21 (e.g., temperature, humidity) | 2020 |
| ETTh1 | Informer(Zhou et al., 2021) | Hourly | 7 states of a electrical transformer | 2016-2017 |
| ETTh2 | Informer(Zhou et al., 2021) | Hourly | 7 states of a electrical transformer | 2017-2018 |
| ETTm1 | Informer(Zhou et al., 2021) | Every 15 minutes | 7 states of a electrical transformer | 2016-2017 |
| ETTm2 | Informer(Zhou et al., 2021) | Every 15 minutes | 7 states of a electrical transformer | 2017-2018 |
| Electricity | UCI ML Repository | Hourly | 321 clients' consumption | 2012-2014 |
| Traffic | Informer(Zhou et al., 2021) | Hourly | 862 sensors' occupancy | 2015-2016 |

---

**Algorithm 3** Reversible Instance Normalization (Revin)

---

1: **Input:** Time-series data $X$, Forecasting model $\mathcal{F}$
2: **Output:** Forecasted data $\hat{X}$
3: **for** each instance $X_i$ in $X$ **do**
4:     Compute mean $\mu_i \leftarrow \text{mean}(X_i)$
5:     Compute variance $\sigma_i^2 \leftarrow \text{variance}(X_i)$
6:     Normalize $\tilde{X}_i \leftarrow \frac{X_i - \mu_i}{\sigma_i}$
7:     **Store** $\mu_i$ and $\sigma_i^2$
8: **end for**
9: $\tilde{X} \leftarrow \{\tilde{X}_1, \tilde{X}_2, \ldots, \tilde{X}_N\}$
10: $\tilde{Y} \leftarrow \mathcal{F}(\tilde{X})$
11: **for** each forecasted instance $\tilde{Y}_i$ **do**
12:     Reverse Normalize $Y_i \leftarrow \tilde{Y}_i \times \sigma_i + \mu_i$
13:     Apply learnable parameters $Y_i \leftarrow \gamma \times Y_i + \beta$
14: **end for**
15: **return** $\hat{X} = \{Y_1, Y_2, \ldots, Y_N\}$

---

reverses the normalization to model outputs by using the saved statistics and applies learnable scaling and shifting parameters ($\gamma$ and $\beta$).

**Sequential Adaptive Normalization.** Sequential Adaptive Normalization (SAN) has two train phases. In the first phase, SAN is trained to learn the relationships between patches of input and target data by mapping their means and variances. In the second phase, SAN parameters are frozen, and only the forecasting model is trained. During inference, input data is normalized using SAN, and the model output is reverse-normalized with predicted statistics by SAN.

### B.3 DETAILS OF THE BACKBONES AND SETUP

In our study, we selected three distinct forecasting models to evaluate the effectiveness of our proposed normalization techniques. DLinear is an MLP-based model renowned for its lightweight architecture, utilizing two separate multilayer perceptrons (MLPs) to learn the periodic and trend components of the data independently.

PatchTST and iTransformer are both Transformer-based models with unique approaches to handling time-series data. PatchTST introduces a patching operation that segments each input time series into multiple patches, which are then used as input tokens for the transformer, effectively capturing local temporal patterns. In contrast, iTransformer emphasizes channel-wise attention by treating the entire sequence of each channel as a transformer token and employing self-attention mechanisms to learn the relationships between different channels.

For all models, we first compute the frequency stability measure across the entire training dataset, a fixed computational process that typically takes less than one second. Following this, we apply a simple, parameter-free normalization and denormalization method. After normalization, the input data is processed through our custom weighting layer before being fed into the forecasting models.

---

**Algorithm 4** Sequential Adaptive Normalization (SAN)

---

1: **Stage 1: Train SAN**
2: **Input:** Training data $X$ and targets $Y$
3: Divide $X$ and $Y$ into patches $\{X_p\}$ and $\{Y_p\}$
4: **for** each pair of patches $(X_p, Y_p)$ **do**
5:     Compute means $\mu_X \leftarrow \text{mean}(X_p)$, $\mu_Y \leftarrow \text{mean}(Y_p)$
6:     Compute variances $\sigma_X^2 \leftarrow \text{variance}(X_p)$, $\sigma_Y^2 \leftarrow \text{variance}(Y_p)$
7:     Train SAN to map $(\mu_X, \sigma_X^2)$ to $(\mu_Y, \sigma_Y^2)$ using loss on $\mu_Y$ and $\sigma_Y^2$
8: **end for**
9: **Stage 2: Train Forecasting Model**
10: Freeze SAN parameters
11: **for** each training iteration **do**
12:     Divide input $X$ into patches $\{X_p\}$
13:     **for** each patch $X_p$ **do**
14:         Normalize $X_p \leftarrow \frac{X_p - \mu_X}{\sigma_X}$ using SAN's learned $\mu_X$ and $\sigma_X^2$
15:     **end for**
16:     Forecast $\tilde{Y} \leftarrow \mathcal{F}(X)$
17:     Divide $\tilde{Y}$ into patches $\{\tilde{Y}_p\}$
18:     **for** each forecasted patch $\tilde{Y}_p$ **do**
19:         Predict $\mu_Y, \sigma_Y^2$ using SAN
20:         Reverse Normalize $Y_p \leftarrow \tilde{Y}_p \times \sigma_Y + \mu_Y$
21:     **end for**
22:     Compute loss $\mathcal{L}(Y, \hat{Y})$
23:     Update forecasting model parameters $\theta$ via backpropagation
24: **end for**
25: **return** Trained forecasting model $\mathcal{F}$

---

### B.4 OTHER EXPERIMENTS DETAILS

**Loss Function.** For our experiments, we adhere to a conventional approach by employing the Mean Squared Error (MSE) loss function, implemented as `nn.MSELoss` in our framework. The MSE loss quantifies the average squared difference between the predicted values and the actual target values, providing a straightforward measure of prediction accuracy. Mathematically, the MSE loss is expressed as $\mathcal{L}_{\text{MSE}} = \frac{1}{N} \sum_{i=1}^{N} (\hat{y}_i - y_i)^2$, where $N$ is the number of samples, $\hat{y}_i$ represents the predicted value, and $y_i$ denotes the true target value for the $i$-th sample. This loss function effectively penalizes larger errors more heavily, encouraging the model to achieve higher precision in its predictions.

**Computational Resources.** All experiments were conducted on an NVIDIA RTX A6000 GPU with 48GB of memory, utilizing CUDA version 12.4 for accelerated computation. This high-performance computational setup facilitated efficient training and evaluation of our forecasting models, ensuring timely execution of experiments even with large-scale time-series data.

## C THE FULL RESULTS.

### C.1 FULL LONG-TERM FORECASTING RESULTS.

Table C.1 presents the comprehensive results discussed in Section 4.2 of our paper. This table includes the prediction accuracy outcomes on the [Electricity, ETTh1, ETTh2, ETTm1, ETTm2, Traffic, Weather] dataset, utilizing the [DLinear, PatchTST, iTransformer] as the backbone model. We have compared our method against all baseline models across all forecasting horizons ($H \in \{96, 192, 336, 720\}$).

### C.2 FULL RESULTS OF RUNNING TIME

Table 7 presents the comprehensive results discussed in Section 4.2 of our paper. This table includes the prediction accuracy outcomes on the [Electricity, ETTh1, ETTh2, ETTm1, ETTm2, Traffic,

Table 6: Detailed results of comparing our proposal and other normalization methods. The best results are highlighted in **bold**.

| Models | | DLinear (Zeng et al., 2023) | | | | | | | | PatchTST (Nie et al., 2023) | | | | | | | | iTransformer (Liu et al., 2024b) | | | | | | | |
|---|---|---|---|---|---|---|---|---|---|---|---|---|---|---|---|---|---|---|---|---|---|---|---|---|---|
| Methods | | + Ours* | | + Ours | | + SAN | | + RevIN | | + Ours* | | + Ours | | + SAN | | + RevIN | | + Ours* | | + Ours | | + SAN | | + RevIN | |
| Metric | | MSE | MAE | MSE | MAE | MSE | MAE | MSE | MAE | MSE | MAE | MSE | MAE | MSE | MAE | MSE | MAE | MSE | MAE | MSE | MAE | MSE | MAE | MSE | MAE |
| Electricity | 96 | **0.135** | **0.230** | 0.140 | 0.237 | 0.137 | 0.234 | 0.210 | 0.278 | 0.175 | 0.266 | 0.190 | 0.280 | 0.182 | 0.271 | 0.212 | 0.297 | 0.145 | 0.244 | **0.143** | **0.237** | 0.171 | 0.262 | 0.152 | 0.251 |
| | 192 | **0.149** | **0.245** | 0.155 | 0.249 | 0.151 | 0.247 | 0.210 | 0.304 | **0.183** | **0.273** | 0.195 | 0.286 | 0.186 | 0.276 | 0.213 | 0.300 | 0.169 | 0.266 | 0.159 | 0.252 | 0.180 | 0.270 | 0.165 | 0.255 |
| | 336 | **0.165** | **0.262** | 0.171 | 0.267 | 0.166 | 0.264 | 0.223 | 0.309 | **0.198** | **0.289** | 0.211 | 0.301 | 0.200 | 0.290 | 0.227 | 0.314 | 0.178 | 0.271 | 0.172 | 0.266 | 0.194 | 0.284 | 0.180 | 0.272 |
| | 720 | **0.198** | **0.291** | 0.208 | 0.298 | 0.201 | 0.295 | 0.257 | 0.349 | **0.233** | **0.317** | 0.253 | 0.334 | 0.237 | 0.322 | 0.268 | 0.344 | 0.210 | 0.311 | 0.205 | 0.295 | 0.237 | 0.319 | 0.227 | 0.312 |
| ETTh1 | 96 | 0.375 | 0.398 | 0.371 | 0.392 | 0.383 | 0.399 | 0.396 | 0.410 | 0.380 | 0.401 | 0.374 | 0.395 | 0.387 | 0.405 | 0.392 | 0.413 | **0.380** | **0.400** | 0.389 | 0.404 | 0.398 | 0.411 | 0.394 | 0.409 |
| | 192 | 0.410 | 0.417 | **0.404** | **0.412** | 0.419 | 0.419 | 0.445 | 0.440 | 0.442 | 0.439 | **0.424** | **0.428** | 0.445 | 0.440 | 0.448 | 0.440 | **0.429** | **0.427** | 0.447 | 0.440 | 0.438 | 0.435 | 0.460 | 0.449 |
| | 336 | 0.430 | 0.427 | 0.426 | 0.426 | 0.437 | 0.432 | 0.487 | 0.465 | 0.480 | 0.456 | 0.471 | 0.452 | 0.505 | 0.471 | 0.489 | 0.456 | 0.479 | 0.451 | 0.492 | 0.463 | 0.481 | 0.456 | 0.501 | 0.475 |
| | 720 | 0.437 | 0.455 | **0.428** | **0.448** | 0.446 | 0.459 | 0.512 | 0.510 | 0.519 | 0.501 | **0.514** | **0.500** | 0.527 | 0.507 | 0.525 | 0.503 | 0.491 | 0.471 | 0.496 | 0.482 | 0.528 | 0.502 | 0.521 | 0.504 |
| ETTh2 | 96 | 0.273 | 0.335 | 0.273 | 0.336 | 0.277 | 0.338 | 0.344 | 0.397 | 0.292 | 0.347 | 0.301 | 0.349 | 0.314 | 0.361 | 0.344 | 0.397 | 0.298 | 0.352 | **0.297** | **0.345** | 0.302 | 0.354 | 0.300 | 0.349 |
| | 192 | 0.335 | 0.374 | 0.336 | 0.376 | 0.340 | 0.378 | 0.485 | 0.481 | 0.385 | 0.402 | 0.380 | 0.399 | 0.391 | 0.421 | 0.389 | 0.411 | 0.371 | 0.402 | 0.380 | 0.395 | 0.383 | 0.402 | 0.381 | 0.415 |
| | 336 | 0.361 | 0.399 | **0.355** | **0.395** | 0.356 | 0.398 | 0.582 | 0.536 | 0.431 | 0.438 | 0.410 | 0.424 | 0.444 | 0.466 | 0.437 | 0.451 | 0.425 | 0.435 | 0.420 | 0.428 | 0.435 | 0.441 | 0.433 | 0.442 |
| | 720 | 0.388 | 0.429 | **0.384** | **0.423** | 0.396 | 0.435 | 0.836 | 0.659 | 0.429 | 0.461 | 0.422 | 0.443 | 0.467 | 0.484 | 0.430 | 0.481 | 0.420 | 0.444 | 0.410 | 0.432 | 0.448 | 0.457 | 0.426 | 0.445 |
| ETTm1 | 96 | 0.285 | 0.339 | 0.299 | 0.341 | 0.288 | 0.342 | 0.353 | 0.374 | 0.322 | 0.359 | 0.321 | 0.362 | 0.325 | 0.361 | 0.353 | 0.374 | 0.326 | 0.361 | 0.330 | 0.370 | 0.331 | 0.373 | 0.341 | 0.376 |
| | 192 | 0.321 | 0.359 | 0.336 | 0.364 | 0.323 | 0.363 | 0.391 | 0.392 | 0.350 | 0.379 | 0.365 | 0.388 | 0.355 | 0.381 | 0.391 | 0.401 | 0.365 | 0.384 | 0.374 | 0.391 | 0.376 | 0.381 | 0.380 | 0.394 |
| | 336 | 0.355 | 0.380 | 0.370 | 0.383 | 0.357 | 0.384 | 0.423 | 0.413 | 0.381 | 0.401 | 0.407 | 0.408 | 0.385 | 0.402 | 0.423 | 0.413 | 0.395 | 0.403 | 0.408 | 0.414 | 0.412 | 0.418 | 0.419 | 0.418 |
| | 720 | 0.405 | 0.411 | 0.425 | 0.414 | 0.409 | 0.415 | 0.486 | 0.449 | 0.446 | 0.436 | 0.464 | 0.442 | 0.450 | 0.437 | 0.486 | 0.459 | 0.471 | 0.447 | 0.475 | 0.449 | 0.485 | 0.453 | 0.486 | 0.455 |
| ETTm2 | 96 | 0.163 | 0.255 | 0.165 | 0.254 | 0.166 | 0.258 | 0.194 | 0.293 | 0.177 | 0.272 | 0.179 | 0.262 | 0.184 | 0.277 | 0.185 | 0.272 | 0.178 | 0.272 | **0.176** | **0.258** | 0.180 | 0.272 | 0.200 | 0.281 |
| | 192 | 0.222 | 0.300 | 0.220 | 0.291 | 0.223 | 0.302 | 0.283 | 0.360 | 0.245 | 0.319 | 0.240 | 0.300 | 0.249 | 0.325 | 0.252 | 0.320 | 0.247 | 0.311 | 0.241 | 0.302 | 0.248 | 0.315 | 0.252 | 0.312 |
| | 336 | 0.272 | 0.329 | 0.273 | 0.325 | 0.272 | 0.331 | 0.371 | 0.450 | 0.298 | 0.253 | 0.310 | 0.347 | 0.330 | 0.378 | 0.315 | 0.351 | 0.307 | 0.351 | 0.307 | 0.347 | 0.308 | 0.352 | 0.314 | 0.352 |
| | 720 | 0.365 | 0.383 | 0.368 | 0.383 | 0.380 | 0.384 | 0.555 | 0.509 | 0.405 | 0.401 | 0.409 | 0.404 | 0.423 | 0.431 | 0.415 | 0.408 | 0.409 | 0.403 | 0.410 | 0.402 | 0.412 | 0.407 | 0.411 | 0.405 |
| Traffic | 96 | 0.410 | 0.286 | **0.408** | **0.277** | 0.412 | 0.288 | 0.648 | 0.396 | 0.497 | 0.342 | 0.527 | 0.339 | 0.530 | 0.340 | 0.650 | 0.396 | **0.400** | **0.271** | 0.394 | 0.268 | 0.502 | 0.329 | 0.401 | 0.277 |
| | 192 | 0.427 | 0.288 | 0.422 | 0.283 | 0.429 | 0.297 | 0.598 | 0.370 | 0.499 | 0.339 | 0.502 | 0.331 | 0.516 | 0.338 | 0.597 | 0.359 | 0.470 | 0.319 | 0.413 | 0.277 | 0.490 | 0.331 | 0.421 | 0.282 |
| | 336 | 0.439 | 0.305 | **0.436** | **0.295** | 0.445 | 0.306 | 0.605 | 0.373 | 0.520 | 0.349 | 0.510 | 0.327 | 0.533 | 0.343 | 0.605 | 0.362 | 0.489 | 0.333 | 0.428 | 0.283 | 0.512 | 0.341 | 0.434 | 0.389 |
| | 720 | 0.454 | 0.311 | 0.455 | 0.311 | 0.474 | 0.319 | 0.645 | 0.395 | 0.550 | 0.349 | 0.545 | 0.345 | 0.575 | 0.367 | 0.642 | 0.381 | 0.478 | 0.330 | 0.463 | 0.301 | 0.576 | 0.364 | 0.465 | 0.302 |
| Weather | 96 | 0.150 | 0.208 | 0.162 | 0.212 | 0.152 | 0.210 | 0.196 | 0.256 | 0.167 | 0.225 | 0.166 | 0.225 | 0.170 | 0.229 | 0.195 | 0.235 | 0.165 | 0.221 | **0.162** | **0.204** | 0.170 | 0.227 | 0.175 | 0.225 |
| | 192 | 0.194 | 0.251 | 0.207 | 0.251 | 0.196 | 0.254 | 0.238 | 0.299 | 0.208 | 0.263 | 0.216 | 0.253 | 0.211 | 0.270 | 0.240 | 0.270 | 0.212 | 0.261 | 0.213 | 0.252 | 0.214 | 0.270 | 0.225 | 0.257 |
| | 336 | 0.243 | 0.289 | 0.256 | 0.288 | 0.246 | 0.294 | 0.281 | 0.330 | 0.255 | 0.301 | 0.273 | 0.295 | 0.261 | 0.310 | 0.291 | 0.306 | 0.261 | 0.304 | 0.271 | 0.295 | 0.265 | 0.309 | 0.280 | 0.307 |
| | 720 | **0.311** | **0.339** | 0.325 | 0.337 | 0.315 | 0.346 | 0.346 | 0.384 | 0.326 | 0.349 | 0.351 | 0.346 | 0.332 | 0.359 | 0.364 | 0.353 | 0.338 | 0.345 | 0.340 | 0.347 | 0.342 | 0.358 | 0.373 | 0.366 |

Table 7: Running time comparison with forecasting lengths $H \in \{96, 720\}$ for all datasets and fixed input sequence length $L = 96$. The **best** results are highlighted.

| Model | Electricity | | ETTh1 | | ETTh2 | | ETTm1 | | ETTm2 | | Traffic | | Weather | |
|---|---|---|---|---|---|---|---|---|---|---|---|---|---|---|
| H | 96 | 720 | 96 | 720 | 96 | 720 | 96 | 720 | 96 | 720 | 96 | 720 | 96 | 720 |
| | DLinear (Zeng et al., 2023) | | | | | | | | | | | | | |
| + Ours | **16.718** | **27.545** | **3.004** | **3.082** | **2.820** | **3.222** | **13.456** | **13.124** | **11.839** | **12.949** | **23.217** | **43.413** | **13.970** | **16.536** |
| + SAN | 19.159 | 34.914 | 8.688 | 8.054 | 8.373 | 7.811 | 36.706 | 36.564 | 36.620 | 36.550 | 31.378 | 54.518 | 37.739 | 40.731 |
| IMP(%) | 12.8% | 21.1% | 65.5% | 60.5% | 66.6% | 58.3% | 63.1% | 64.2% | 67.8% | 64.3% | 26.0% | 20.3% | 63.0% | 59.9% |
| | PatchTST (Nie et al., 2023) | | | | | | | | | | | | | |
| + Ours | **99.104** | **103.781** | **7.952** | **8.215** | **14.687** | **14.122** | **54.526** | **28.944** | **59.406** | **26.233** | **209.006** | **215.718** | **69.107** | **49.628** |
| + SAN | 313.697 | 322.083 | 16.226 | 15.309 | 16.078 | 14.998 | 63.686 | 65.839 | 65.978 | 66.798 | 550.026 | 557.730 | 81.973 | 81.462 |
| IMP(%) | 68.4% | 67.7% | 51.6% | 46.2% | 8.00% | 5.20% | 14.3% | 56.0% | 10.0% | 60.7% | 61.6% | 61.2% | 15.9% | 39.8% |

Weather] dataset, utilizing the [DLinear, PatchTST] as the backbone model. We have compared our method against all baseline models across all forecasting horizons ($H \in \{96, 720\}$).

