# OpenReview forum: "FredNormer: Frequency Domain Normalization for Non-stationary Time Series Forecasting"
_ICLR.cc/2025/Conference — ICLR 2025 Conference Withdrawn Submission_

### Official Review · Reviewer_s19A · 2024-10-27

**Soundness:** 2
**Presentation:** 3
**Contribution:** 3
**Rating:** 5
**Confidence:** 5

**Summary:**

This research tackles a fundamental challenge in time series forecasting—the non-stationarity of real-world data—by introducing a novel perspective on normalization within the frequency domain. The authors propose a versatile plug-and-play module that analyzes datasets from a frequency standpoint and dynamically enhances the significance of key frequency components. Comprehensive experiments demonstrate that the FredNormer module enhances the average Mean Squared Error (MSE) of various underlying forecasting models.

**Strengths:**

The paper is generally well-structured, with a clear progression from introduction to conclusion. The use of technical language is appropriate for the audience, and the figures and tables are mostly supportive of the text.

The authors propose FredNormer, a frequency domain normalization method that adaptively up-weights key frequency components based on their stability. The method is model-agnostic and can be integrated into various forecasting models without compromising efficiency.

This paper tackles an important problem in time series forecasting—the non-stationarity of real-world data. The idea of approaching normalization from the frequency domain is novel and could potentially offer new insights into the field.

**Weaknesses:**

The importance of the questions posed by the paper is not in doubt, but the execution and resulting conclusions may be less reliable due to experimental concerns. The technical claims of the paper are where the most significant concerns lie. The experimental methodology appears to have several flaws, which may compromise the validity of the results. Without a solid experimental foundation, the technical claims, no matter how innovative, cannot be adequately supported. The paper would benefit from a thorough revision of the experimental design and possibly additional experiments to validate the findings.

The initial discussion in the article pertains to the equivalence of linear combinations before and after applying the Fast Fourier Transform (FFT). However, this proof fails to address the first-order differences introduced in the practical algorithmic procedure. Specifically, Algorithm 1 computes a quantity $S$ based on the batch mean and variance of frequency domain points derived from the input sequence. Conversely, Algorithm 2 utilizes this $S$ to perform a linear transformation on the frequency domain representation of the first-order differences of the sequence. It is evident that the actual algorithmic workflow does not fully align with the theoretical proof provided.

When evaluating the experimental runtime, it is important to note that if the speed is not measured during the stable phase of training, factors such as initial data loading during training runs may obscure the speed reduction effect attributable to the FredNormer plugin.

The potential contribution of the paper to the field is significant if the method can be validated. However, as it stands, the experimental issues limit the trust that can be placed in the results.

**Questions:**

1. Why is there a lack of discussion and explanation on the introduction of the first-order difference in the paper, especially when it does not align with the proofs provided for Algorithm 1 and Algorithm 2?

2. In the Frequency Stability Measure Analysis, why are there no comparisons with other methods in the ablation experiments, specifically when the proposed FredNormer method is being evaluated?

3. The experimental results for the reproduction of iTransformer and PatchTST in the current paper appear significantly inferior to those reported in their original papers. When comparing the FredFormer's experimental results to those in other papers, there seems to be no notable improvement.

Liu, Y., Hu, T., Zhang, H., Wu, H., Wang, S., Ma, L., & Long, M. (2023). itransformer: Inverted transformers are effective for time series forecasting. arXiv preprint arXiv:2310.06625.
Nie, Y., Nguyen, N. H., Sinthong, P., & Kalagnanam, J. (2022). A time series is worth 64 words: Long-term forecasting with transformers. arXiv preprint arXiv:2211.14730.

---

### Official Review · Reviewer_8Dbc · 2024-11-03

**Soundness:** 3
**Presentation:** 3
**Contribution:** 2
**Rating:** 5
**Confidence:** 4

**Summary:**

This paper presents FredNormer, a method designed to tackle the distribution shift issue in the frequency domain. It comprises two components: a statistical metric and a learnable weighting layer. Extensive experiments demonstrate that FredNormer enhances forecasting performance.

**Strengths:**

1. This paper is well-written and easy to follow.

2. Extensive experiments have demonstrated that FredNormer enhances the performance of backbone forecasting models, resulting in notable improvements.

**Weaknesses:**

My main concern is regarding the contribution of this work. What are the key differences between FredNormer and the two referenced papers [1][2]?



[1] Frequency Adaptive Normalization For Non-stationary Time Series Forecasting, in NeurIPS 2024


[2] Deep Frequency Derivative Learning for Non-stationary Time Series Forecasting, in IJCAI 2024

**Questions:**

please refer to the weakness

---

### Official Review · Reviewer_NuKM · 2024-11-04

**Soundness:** 2
**Presentation:** 2
**Contribution:** 2
**Rating:** 3
**Confidence:** 3

**Summary:**

The paper notes that normalization-based methods can only partially address the distribution shift problem in non-stationary time series forecasting, as these methods operate solely in the time domain and may overlook prominent dynamic patterns in the frequency domain. To tackle this, they propose FredNormer, a plug-and-play module that dynamically adjusts the weight of each frequency component based on a proposed measure termed Frequency Stability $\mathrm{S}$.

**Strengths:**

* The authors define Frequency Stability $\mathrm{S}$ across the entire dataset based on the discrete Fourier transform coefficients, and they theoretically demonstrate that time-domain normalization can only uniformly scale non-zero frequency components, leaving the proportion of the Stable Frequency Subset $\mathcal{O}$ within the spectrum unchanged after normalization.

* The authors propose to enhance stable frequency components for better generalization in TS forecasting. They introduce a learnable layer that assigns greater weights to stable frequency elements based on $\mathrm{S}$.

**Weaknesses:**

* The contribution of the paper seems limited. According to equation (10), the core approach mainly involves applying a linear transformation to frequency components based on the matrix $\mathrm{S}$. It is unclear how $\mathrm{S}$ functions as a constraint to realize the key motivation of enhancing the weight of stable frequency components.

* In Definition 2, the authors introduce the concept of a Stable Frequency Subset $\mathcal{O}$ but do not provide a clear criterion for determining what qualifies as a stable frequency. Some notations, such as Stable Frequency Subset $\mathcal{O}$, and Theorem 1 presented in Section 2 are not well linked to the design of the method proposed in Section 3.

* The experimental evaluation lacks thoroughness, as it includes only three baseline models (Dlinear, PatchTST, and iTransformer). Although the authors mention other models, such as CrossFormer, they do not include it in their comparisons. Additionally, the Nonstationary Transformer [1] would also be a relevant model for further comparison.

* The empirical analysis in Figure 3 is confusing to the reviewer. Could the authors provide additional clarification on how the adjustments to the amplitudes of the input series and forecasting target, based on the Frequency Stability score, affect model prediction accuracy? Additionally, could you explain why these adjustments are effective in enhancing the model's performance? Both Equations (9) and (10) have large spacing from the preceding text.

[1] Liu, Yong, et al. "Non-stationary transformers: Exploring the stationarity in time series forecasting." Advances in Neural Information Processing Systems 35 (2022): 9881-9893.

**Questions:**

* In the remark on line 196, the authors note that intermixed stable and unstable components lead to entangled patterns and thus lead to sub-optimal forecasting performance of current models. Could the authors clarify how their proposed model differentiates between stable and unstable components?

* What is the purpose of smoothing the data before applying the discrete Fourier transform in Equation (8)? How might the performance of the proposed model be impacted if the transform were applied directly to the raw data?

Also, see the weaknesses.

---

### Official Review · Reviewer_LfuW · 2024-11-04

**Soundness:** 3
**Presentation:** 3
**Contribution:** 2
**Rating:** 5
**Confidence:** 4

**Summary:**

This paper focuses on the issue of distribution shift in the frequency domain to improve non-stationary time series forecasting. Its primary contribution lies in the theoretical analysis of how normalization methods impact frequency components and in adaptively up-weighting key frequency components. Specifically, the proposed method, FredNormer, includes a statistical metric to normalize input samples and a learnable weighting layer to adjust stability.

**Strengths:**

1. The authors provide a theoretical proof showing that time-domain normalization is not effective for frequency components.
2. The proposed method is a simple and effective method for learning time-invariant frequency components to suppress non-stationary in the frequency domain.
3. FredNormer is demonstrated to be effective, achieving state-of-the-art (SOTA) performance in experiments.

**Weaknesses:**

1. FAN is also a method based on adaptive normalization in the frequency domain, capable of handling both trend and seasonal non-stationary patterns in time series data. Compared with it, although this paper provides a theoretical guarantee, its contribution appears insufficient.

[1] Ye W, Deng S, Zou Q, et al. Frequency Adaptive Normalization For Non-stationary Time Series Forecasting[J]. Advances in Neural Information Processing Systems, 2024.

2. In Non-stationary Transformer, the authors argue that removing inherent non-stationarity from time series may reduce the model's ability to forecast real-world bursty events. Could suppressing non-stationary information in FredNormer similarly lead to over-stationarization, limiting its practical applicability?

[1] Liu Y, Wu H, Wang J, et al. Non-stationary transformers: Exploring the stationarity in time series forecasting[J]. Advances in Neural Information Processing Systems, 2022, 35: 9881-9893.

3. In Definition 2, you mention "M components with higher stability S(k)." How do you define "higher" stability? What is the threshold value used, and how is it selected?

4. The authors only compare against with Transformer-based and MLP-based methods. Why not include comparisons with more CNN-based baselines, such as TSLANet, ModernTCN, and TimesNet, to demonstrate the robustness and generalization of FredNormer?

[1] Eldele E, Ragab M, Chen Z, et al. TSLANet: Rethinking Transformers for Time Series Representation Learning[C]//Forty-first International Conference on Machine Learning, 2024.

[2] Luo D, Wang X. Moderntcn: A modern pure convolution structure for general time series analysis[C]//The Twelfth International Conference on Learning Representations. 2024.

[3] Wu H, Hu T, Liu Y, et al. TimesNet: Temporal 2D-Variation Modeling for General Time Series Analysis[C]//The Eleventh International Conference on Learning Representations, 2023.

5. I am curious about FredNormer’s impact on frequency-domain methods like FiTS and FreTS. Have you tested its performance on these models?

Minor Error:
1. Section 4.2 mentions running time, but Figure 4 seems to be missing.
2. In Section 4.2, you refer to a "fourth-shaped frequency component." Could you clarify what you mean by this?
3. In the notations section, the indicator function I{k=0} is said to equal 1 if k is not equal to 0.
In Proof 1, you state that for k≠1,I{k=0}=0. Could you clarify which case is correct?

**Questions:**

1. Regarding the contribution, refer to W1.

2. For the experiment, please see W4 and W5.

---

### Official Review · Reviewer_xnnx · 2024-11-04

**Soundness:** 2
**Presentation:** 2
**Contribution:** 2
**Rating:** 3
**Confidence:** 5

**Summary:**

This paper reveals that traditional time-domain normalization methods uniformly scale non-zero frequencies, which limits their ability to effectively handle distribution shifts in time series forecasting. To address this limitation, the authors propose FredNormer, a plug-and-play module that combines statistical frequency stability normalization with learnable sample-specific weighting, enabling better adaptation to key frequency components and more robust forecasting performance.

**Strengths:**

1. this paper studies an interesting problem in time series forecasting, i.e., the distribution shift problem.

2. the paper is understandable and easy to follow.

**Weaknesses:**

1. this paper is not very well-motivated. The paper mentioned that "Modeling solely in the time domain struggles to distinguish between different frequency components within superimposed time series" as their first motivation. Does it have any difference with the distribution shift or non-stationary forecasting? Also, it mentioned that "the z-score normalization applies uniform scaling across all frequency components, which leaves frequency-specific patterns unaltered". I believe only RevIN uses "learnable" z-score normalization but other methods like SIN, SAN, FAN do not use z-score normalization. What they do cannot be equal to z-score normalization. So how can you use z-score to summarize these works?

2. this paper is kind of confusing for the theoretical analysis. Based on the Theorem 1, the authors mentioned that "the normalization operation keeps the proportion unchanged". The proportion is unchanged does not mean it cannot normalize the time series. So the theoretical analysis cannot show the existing normalization would fail. Moreover, how can you say your defined stability is correlated with the non-sationarity? Why the frequency components are entangled and thus influence normalization? These parts don't have theoretical analysis I believe.

3. the experiments are somehow lack of comparisons with state-of-the-art normalization techniques, such as SIN [1].

[1] SIN: Selective and Interpretable Normalization for Long-Term Time Series Forecasting. In ICML.

**Questions:**

See weakness.

---

### Note · Authors · 2024-11-17

**Comment:**

We sincerely thank the reviewers for their valuable comments and insights. After thorough consideration, we have decided to withdraw our manuscript.

**Withdrawal Confirmation:**

I have read and agree with the venue's withdrawal policy on behalf of myself and my co-authors.